# Risk-Adapted Treatment Strategies with Pre-Irradiation Chemotherapy in Pediatric Medulloblastoma: Outcomes from the Polish Pediatric Neuro-Oncology Group

**DOI:** 10.3390/children10081387

**Published:** 2023-08-15

**Authors:** Marta Perek-Polnik, Anne Cochrane, Jinli Wang, Marzanna Chojnacka, Monika Drogosiewicz, Iwona Filipek, Ewa Swieszkowska, Magdalena Tarasinska, Wiesława Grajkowska, Joanna Trubicka, Paweł Kowalczyk, Bożenna Dembowska-Bagińskai, Mohamed S. Abdelbaki

**Affiliations:** 1Department of Oncology, Children’s Memorial Health Institute, 01-211 Warsaw, Poland; m.drogosiewicz@ipczd.pl (M.D.); i.filipek@ipczd.pl (I.F.); e.swieszkowska@ipczd.pl (E.S.); m.tarasinska@ipczd.pl (M.T.); b.dembowska@ipczd.pl (B.D.-B.); 2Division of Hematology and Oncology, Department of Pediatrics, School of Medicine, Washington University, St. Louis, MO 63110, USA; 3Center for Biostatistics and Data Science, Washington University School of Medicine, St. Louis, MO 63110, USA; 4Maria Sklodowska-Curie National Research Institute of Oncology, Pediatric Radiotherapy Centre, 00-001 Warsaw, Poland; 5Department of Pathology, Children’s Memorial Health Institute, 01-211 Warsaw, Poland; w.grajkowska@ipczd.pl (W.G.); j.trubicka@ipczd.pl (J.T.); 6Department of Neurosurgery, Children’s Memorial Health Institute, 01-211 Warsaw, Poland; p.kowalczyk@ipczd.pl

**Keywords:** medulloblastoma, chemotherapy, radiotherapy, pediatrics, brain tumors

## Abstract

Craniospinal irradiation (CSI) has been a major component of the standard of care treatment backbone for childhood medulloblastoma. However, chemotherapy regimens have varied based on protocol, patient age, and molecular subtyping. In one of the largest studies to date, we analyzed treatment outcomes in children with newly-diagnosed medulloblastoma treated with pre-irradiation chemotherapy followed by risk-adapted radiotherapy and maintenance chemotherapy. A total of 153 patients from the Polish Pediatric Neuro-Oncology Group were included in the analysis. The median age at diagnosis was 8.0 years, and median follow-up time was 6.4 years. Sixty-seven patients were classified as standard-risk and eighty-six as high-risk. Overall survival (OS) and event-free survival (EFS) for standard-risk patients at 5 years (±standard error) were 87 ± 4.3% and 84 ± 4.6%, respectively, while 5-year OS and EFS for high-risk patients were 81 ± 4.3% and 79 ± 4.5%, respectively. Only one patient had disease progression prior to radiotherapy. This study demonstrates promising survival outcomes in patients treated with pre-irradiation chemotherapy followed by risk-adapted CSI and adjuvant chemotherapy. Such an approach may be useful in cases where the initiation of radiotherapy may need to be delayed, a common occurrence in many institutions globally.

## 1. Introduction

Medulloblastoma is the most common pediatric central nervous system (CNS) malignancy, comprising over 60% of pediatric embryonal tumors. Its peak incidence occurs in patients of 0–9 years of age and declines after age 15 [1]. Medulloblastoma is more predominant in males, affecting approximately 1.5 times more male than female patients [2]. The tumor is most commonly classified into four molecular subtypes: wingless (WNT), sonic hedgehog (SHH), group 3, and group 4 [3,4]. In children younger than three years of age, the predominant subtypes are SHH and group 3, with the SHH group typically associated with more favorable outcomes [5]. In older children and adults, WNT and group 4 are more commonly seen, and the latter is typically associated with poorer prognosis and the presence of metastatic disease at the time of diagnosis [6].

Treatment plans for medulloblastoma are typically dependent on risk-stratification; standard-risk and high-risk groups are classified by several clinical and pathologic risk factors. Standard of care therapy consists of surgical resection, craniospinal irradiation (CSI), and adjuvant chemotherapy with platinum and alkylator-based agents [7]. CSI has been a mainstay of treatment given medulloblastoma’s radiosensitivity and predilection for leptomeningeal disease. These contemporary treatment strategies are typically associated with 5-year event-free survival (EFS) rates of 75–80% among average-risk patients and 60–70% for high-risk patients [7,8].

The adverse neurologic and developmental sequela of CSI have been well-documented in pediatric brain tumor patients [9]. These effects include, but are not limited to, increased risk of secondary malignancy, neurocognitive decline, and endocrine dysfunction [9,10,11]. Late effects such as these ultimately contribute to long-term psychosocial and quality of life impairment in medulloblastoma survivors. With the goal of mitigating the adverse sequela of radiation therapy, attempts at lowering the dose have been explored. In the 1990s, the dose of CSI in standard-risk medulloblastoma patients was safely reduced from 36.0 Gy to 23.4 Gy [12]. Subsequent studies have investigated strategies to further de-intensify or postpone radiotherapy.

More recently, use of adjuvant and neoadjuvant chemotherapy has been proposed to minimize the adverse sequela of CSI by allowing dose-reduction or delayed initiation of radiotherapy [13,14]. Studies incorporating the use of pre-irradiation chemotherapy were initially explored among infants and young children, where delaying radiation therapy is particularly important to preserving neurocognitive development. One of the first studies exploring this approach was the 1986 Pediatric Oncology Group (POG) study treating children under 36 months of age with malignant brain tumors using post-operative chemotherapy to delay radiotherapy until 1–2 years after diagnosis [15]. The dose of irradiation was subsequently reduced in patients who did not have evidence of residual disease following chemotherapy. Furthermore, the SIOP PNET-3 study demonstrated improved 5-year EFS for medulloblastoma patients between the ages of 3 to 16 treated with pre-irradiation chemotherapy compared to those receiving only radiotherapy [14].

Other recent studies investigating deintensification of radiotherapy in older children include ACNS 0331, where reduction of field volume boost in standard-risk patients of 3–21 years of age was not associated with inferior outcomes [16]. The St. Jude medulloblastoma protocols from 1996 and 2003 utilized risk-adapted radiotherapy regimens of 23.4 Gy and 36–39.6 Gy CSI for standard-risk and high-risk patients, respectively, with favorable outcomes [17,18]. Hence, we aim to add to the literature by reporting on the largest cohorts to-date, wherein patients with medulloblastoma were treated with a pre-irradiation chemotherapy approach.

## 2. Methods

### 2.1. Study Design and Population

This study included 153 patients (3–31 years) with newly-diagnosed medulloblastoma who were treated at the Children’s Memorial Health Institute in Warsaw, Poland between January 2010 and September 2021. All patients underwent gadolinium-enhanced brain and spine MRI before and after surgery to assess the disease’s extent. Cerebrospinal fluid sampling was performed on most patients to evaluate for disseminated disease. This was either conducted via ventricular CSF sampling at time of surgery or via lumbar puncture. Diagnoses were confirmed histologically via central pathology review, and specimens were evaluated for large cell/anaplastic features to assist with risk stratification. Molecular analysis was performed on a select number of tumor specimens. Patients were classified post-operatively into standard-risk and high-risk groups per historical risk stratification criteria.

Standard-risk patients showed no evidence of metastatic disease confirmed by gadolinium-enhanced MRI of the head and spine, lumbar cerebrospinal fluid sampling prior or after surgery contained no tumor cells, or if the residual tumor was 1.5 cm^2^ or less following resection. High-risk patients show evidence of metastatic disease confirmed on gadolinium-enhanced MRI in the head and spine, lumbar cerebrospinal fluid (CSF) following resection, large cell/anaplastic histology, or if there was residual tumor greater than 1.5 cm^2^ post-operatively.

This study was conducted in accordance with guidelines established by the local bioethics committee and the Declaration of Helsinki. Informed consent was obtained from parents or legal guardians of all the participants, and all patient information was anonymized.

### 2.2. Treatment Protocol

Maximum safe tumor resection was attempted whenever possible. Patients underwent placement of external ventricular drain or ventriculoperitoneal shunt as deemed appropriate by the institution’s neurosurgical team. The extent of resection was assessed peri-operatively as gross total resection (GTR) if less than 1.5 cm^2^ of residual tumor remained or subtotal resection (STR) if residual tumor was greater than 1.5 cm^2^. Chemotherapy was initiated within two weeks of surgical resection whenever possible.

Standard-risk patients were treated with multimodal regimens adapted from the SIOP PNET-3 trial and Packer 2006 study [13,14]. Induction comprised vincristine, carboplatin, etoposide, and cyclophosphamide prior to irradiation, followed by vincristine, cisplatin, and CCNU (Table 1). High-risk patients were treated with a protocol incorporating the ICE regimen adapted from the Sawamura 1996 study [19]. This consisted of carboplatin, etoposide, vincristine, and ifosfamide prior to irradiation, followed by vincristine, CCNU, and cisplatin for consolidation (Table 2).

Chemotherapy was started within 14 days of surgical resection and confirmation of histopathologic diagnosis of medulloblastoma. The first course of chemotherapy did not differ between the standard and the high-risk group, so the treatment could be started prior to completion of molecular analysis. Hematological requirements to start treatment included an ANC > 1 tys/µL, hemoglobin concentration ≥ 10 mg/dL, and platelet count ≥ 100,000 tys/µL. Hepatic and renal function was also assessed. The requirements included AST and ALT ≤ 2.5 times normal limits for age, creatine level within normal limits for age, and creatine clearance ≥ 60 mL/min. To maintain compliance, 8 rounds of maintenance chemotherapy were given, but intervals between courses could be extended by an additional 2 weeks to allow for blood count recovery to a minimum of ANC 0.75 tys/µL, hemoglobin concentration ≥ 10 mg/dL, and platelet count of 75,000 tys/µL. Since 2013, patients unable to meet these parameters have received an alternative post-radiotherapy treatment course with vincristine and cyclophosphamide alternating with vincristine with cisplatin, omitting lomustine. In those patients, hematological requirements were lowered to a WBC of 1 tys/µL and platelet count of 30,000 tys/µL. Since 2016, patients have received a minimum of 14 doses of vincristine during maintenance chemotherapy. If toxicity attributed to vincristine occurred, the subsequent doses were lowered by half or fully omitted, and the full dose was resumed after symptom resolution. No changes in the treatment were implemented if hearing loss was diagnosed in the patient during treatment.

A risk-adapted radiotherapy regimen was used, where standard-risk patients received a 25 Gy CSI dose with boost to the tumor bed, and high-risk patients received 36 Gy CSI dose with boost to the tumor bed. Patients who had metastatic disease at diagnosis underwent post-chemotherapy evaluation, and those without complete remission received a boost to the site of metastasis. Since 2010, this approach has been the standard of care for medulloblastoma treatment in Poland.

All patients were irradiated with megavoltage X-ray beams via a linear accelerator. Starting in 2014, patients were treated with volumetric arc therapy (VMAT) techniques. Prior to this, intensity-modulated radiotherapy (IMRT) and 3D conformal photon radiotherapy (3DCRT) were used. When planning radiotherapy, all patients had individual thermoplastic five-point fixed masks optimized for head and shoulder immobilization to ensure reproducibility during treatment. While immobilized, a CT scan using 2.5 mm slices was performed. The target volumes and organs at risk were contoured using the simulated CT images along with pre-operative and post-operative MRI scans. The craniospinal clinical target volume (CTV) included the whole brain, meninges, spine, and nerve roots laterally. The caudal component of the CTV comprised the entire subarachnoid space to the lower limit of the thecal sac as visualized on MRI. All treatment plans were standardized to a craniospinal dose of 25.2 Gy (14 fractions, 1.8 Gy per day) for standard-risk patients and 36 Gy (20 fractions, 1.8 Gy per day) for high-risk patients. The radiation dose was Increased for the tumor bed and residual tumor with 1.5 cm margins or greater, with a maximum dose of 55.8 Gy. In the high-risk group, the dose was also increased for metastases. The target dose was determined independently depending on the size, number, and location of metastases along with their response to chemotherapy.

### 2.3. Statistical Analysis

Categorical patient characteristics were reported as frequencies and percentages. Continuous variables were reported as median values with interquartile ranges (IQR). The primary treatment endpoints were EFS and overall survival (OS). EFS was defined as time from diagnosis to progression, disease recurrence, or death. OS was defined as time from diagnosis to death. Subjects without an event during the study period were censored at the date of last follow-up. The Kaplan–Meier method was used to construct the cumulative survival curves and estimate survival rates with SE via Greenwood’s formula. Statistical analysis was performed using SAS software, version 9.4 (SAS Institute).

## 3. Results

### 3.1. Demographics and Patient Characteristics

Baseline characteristics and demographics are listed in Table 3. From an original cohort of 155 patients, 153 were included in the analysis; 2 patients were excluded from the analysis due to incomplete treatment data. The median age at diagnosis was 8.0 years (IQR: 5.4–11.0). Of the patients, 96% were male and 32% female; 67 patients (44%) were classified as standard-risk and 86 (56%) as high-risk at the time of diagnosis. The median follow-up time for the entire patient cohort was 6.4 years (IQR: 3.6–9.2, range: 0.31–13.2). Standard-risk patients had a median follow-up time of 6.8 years (IQR: 3.8–9.9). High-risk patients had a median follow-up time of 6.2 years (IQR: 3.6–9.0). Among high-risk patients, 66% achieved GTR. All 67 standard-risk patients achieved GTR. Fifty four (63%) high-risk patients had evidence of metastasis at time of diagnosis. Large-cell/anaplastic features were seen in 43 (50%) of high-risk patients.

All tissue samples were analyzed for c-Myc and n-Myc amplification via fluorescent in situ hybridization. Starting in 2012, patients underwent analysis for mutations in beta catenin and Ch6 monosomy and were classified to the WNT subgroup if indicated. Likewise, samples with YAP or Gli expression on immunohistochemistry were further evaluated for classification into the SHH subgroup. Patients without Myc amplification were not categorized as a specific subgroup. Seven (10%) standard-risk patients were classified into the WNT subgroup. Four (5%) high-risk patients were classified into the WNT subgroup and two (2%) were classified into the SHH subgroup.

### 3.2. Survival Outcomes

Survival outcomes are described in Table 4 and Figure 1 and Figure 2 below. Of the 67 standard-risk patients, 56 (84%) remained alive at the time of last follow-up; 68 (79%) of 86 high-risk patients were alive at the time of last follow-up. Overall survival (OS) and event-free survival (EFS) for standard-risk patients at 5 years ± standard error (SE) were 87 ± 4.3% and 84 ± 4.6%, respectively, while the 5-year OS and EFS for high-risk patients were 81 ± 4.3% and 79 ± 4.5%, respectively. Only one patient had disease progression prior to radiotherapy. Twelve (18%) standard-risk patients had disease recurrence, eleven of whom died after recurrence. Eighteen (21%) high-risk patients had disease recurrence, all of whom died after recurrence.

## 4. Discussion

This study evaluates the outcomes of medulloblastoma patients receiving pre-irradiation chemotherapy and risk-adapted radiotherapy as part of their treatment regimen. The OS and EFS associated with the standard-risk and high-risk patient population suggests that strategies aimed at reducing tumor burden prior to radiotherapy may lead to improved survival outcomes. It has been proposed that earlier initiation of chemotherapy also allows for better local control prior to radiation therapy and decreased risk of disease recurrence [20]. Since this approach allowed for more post-operative recovery time, patients required less anesthesia interventions while receiving radiotherapy. Other advantages include improved local delivery of chemotherapy in the immediate postoperative period, facilitated by disruption in the blood–brain barrier, and better tolerability of chemotherapy prior to radiation-related bone marrow compromise. [21]. While not applicable to our protocol, studies have demonstrated reduced ototoxicity of platinum-containing agents when administered prior to radiation therapy [22].

Our results are consistent with existing international studies demonstrating favorable survival outcomes in patients treated with pre-irradiation chemotherapy protocols [23]. Many studies utilizing pre-irradiation chemotherapy strategies were targeted at children less than 3 years of age who are more vulnerable to the effects of irradiation. Earlier trials including the German HIT-SKK’89 and HIT-SKK’92 utilized post-surgical chemotherapy and delayed radiotherapy until children reached 3 years of age [24]. The subsequent HIT2000 trial intensified neoadjuvant and adjuvant chemotherapy, yielding improved 5 year EFS of 62%, compared to 47% from HIT92 [25]. A Japanese regimen utilizing pre-irradiation ifosphamide, cisplatin, and etoposide yielded a 5-year OS of 82% for patients younger than 20 years of age with medulloblastoma [26]. The St. Jude Medulloblastoma trials from 1996 and 2003 employing risk-adapted radiotherapy regimens reported 5-year EFS of 82–83% in standard-risk and 60–70% in high-risk patients.

Some studies have demonstrated significant success while administering mainly chemotherapy-only strategies, including a 2005 study by Rutkowski et al. treating patients less than 3 years of age with 6 months of postoperative chemotherapy alone, yielding 5-year OS rates of 66% [27]. Children treated with this regimen also demonstrated improved neurocognitive outcomes, with mean IQ higher than that of patients in a previous trial receiving radiotherapy. The Head Start I and II trials investigated use of intensive myeloablative chemotherapy and autologous hematopoietic progenitor cell rescue (AuHCR) in patients less than 3 years old with non-metastatic medulloblastoma, irradiating patients only if relapse occurred [28]. Using this approach, 71% of patients were able to avoid irradiation. Five-year OS for patients with desmoplastic and classical medulloblastoma were 78% and 67%, respectively. Mean intellectual functioning and quality of life scores for children that did not receive radiation remained within the average range among all survivors.

A common concern in protocols utilizing pre-irradiation chemotherapy is the risk of disease progression in patients prior to radiotherapy [29]. In our study, only one patient had disease progression prior to radiotherapy. Other international studies utilizing pre-irradiation chemotherapy strategies have also reported low rates of disease progression prior to radiotherapy [30]. A 2006 report from Egypt demonstrated no disease progression in a cohort of 20 patients with high-risk medulloblastoma who received two cycles of pre-irradiation chemotherapy over a 5-week duration [23]. In the SIOP PNET-3 trial, only 3 of 179 patients did not progress to radiotherapy, 1 due to parental refusal, 1 due to death from chemotherapy toxicity, and 1 due to extent of disease progression [14]. The study reported overall survival of 79.5% and 70.7% at 3 and 5 years, respectively. Poor outcomes due to the delay in chemotherapy have also been reported; in an Italian study from 1999, 4 out of 12 patients developed progressive disease by the second cycle of pre-radiation chemotherapy [31]. With this in mind, the selection of therapeutic agents and intensity of therapy should be closely considered to minimize the incidence of progressive disease during the pre-radiation phase of treatment.

There is a wide variation in survival outcomes for medulloblastoma patients around the world as described by a recent study by Girardi et al. [32]. Data from low- and middle-income countries yielded much lower survival rates for medulloblastoma than high-income countries. Contributory factors include differences in diagnostic methodologies, heterogeneity in cancer data registries, or lack of access to therapeutic interventions. In light of this, it is important to consider diagnostic challenges and barriers to reliable and cost-effective treatment. One such example is accessibility to molecular characterization of tumors, which can allow for better risk-stratification and targeted therapy for patients, as different subgroups may be responsive to different therapeutic strategies [33,34]. Recent clinical trials including SJMB12 and ACNS1422 have started to explore this approach, utilizing radiotherapy dose-reduction for the more favorable WNT subgroup [17]. These studies remain active and survival outcomes are not yet available for reporting.

Similar to all retrospective studies, additional clinical data is necessary in order to establish definitive treatment conclusions for our patient cohort. Our study is restricted by a limited sample size and data collection from a single institution, which may reflect selection bias. The lack of molecular subtyping in our patient cohort was another limitation of our study, as only a small population of tissue samples had subtyping available at time of data collection. Obtaining molecular subgrouping of more samples would allow for better interpretation of survival outcomes, as differences in treatment response and survival have been described between the different subgroups [35,36]. Studies investigating treatment for high-risk medulloblastoma including ACNS0332 and SJMB03 have reported on treatment response differences based on molecular subgrouping and highlighted discrepancies in histopathologic diagnoses in tumor classification [17,37]. Between the two studies, 5-year EFS ranged between 93 and 98% for patients in the WNT subgroup, 75 and 83% for SHH, 63 and 67% for Group 3, and 86 and 87% for those in Group 4.

The quality of life, endocrinologic, and neurocognitive outcomes of our patients remain to be analyzed. Retrospective and prospective studies investigating long-term outcomes among medulloblastoma patients treated with craniospinal irradiation have reported high rates of late effects, and we anticipate this among our patient cohort as well. The retrospective Institute Curie study examining late effects among pediatric medulloblastoma patients treated between 1980 and 2000 reported neurologic deficits among 71% of patients and endocrine complications among 52% [38]. Long-term outcomes of medulloblastoma survivors remain limited, and additional studies reporting on its late effects will be important to improving psychosocial outcomes and reducing morbidity in this patient population. Future directions include the molecular sequencing of additional tissue samples to allow for improved risk stratification, along with the analysis of morbidity and the late effects of treatment.

## 5. Conclusions

In conclusion, we have reported the results of medulloblastoma patients treated with pre-irradiation chemotherapy and risk-adapted craniospinal radiation. While strategies for delaying radiation therapy have been explored for young children and infants, data on pre-irradiation chemotherapy in older medulloblastoma patients remain limited. The favorable survival data in our patient population indicates that pre-irradiation chemotherapy may allow for improved outcomes in older pediatric medulloblastoma patients. Such an approach may be helpful in countries with limited resources where access to radiotherapy may be delayed.

## Figures and Tables

**Figure 1 children-10-01387-f001:**
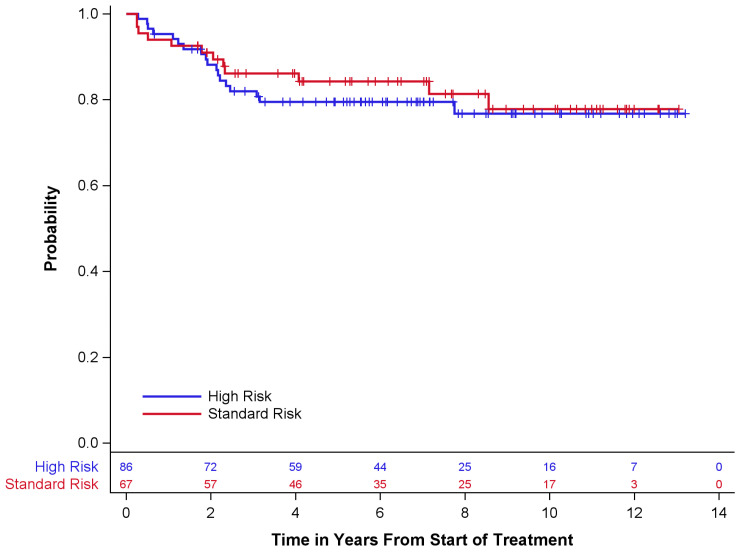
Event-free survival by risk group.

**Figure 2 children-10-01387-f002:**
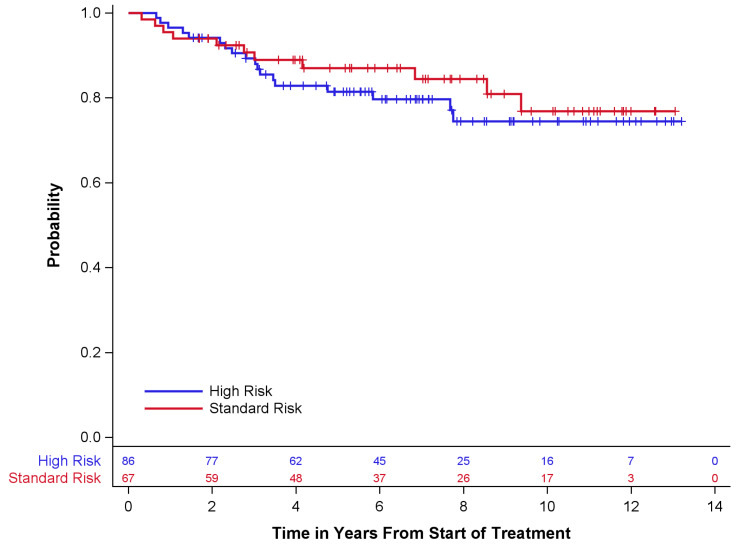
Overall survival by risk group.

**Table 1 children-10-01387-t001:** Standard-risk treatment protocol.

Induction	
Day	Therapy
1, 42	Vincristine 1.5 mg/m^2^
	Carboplatin 500 mg/m^2^
	Etoposide 100 mg/m^2^
2, 43	Carboplatin 500 mg/m^2^
	Etoposide 100 mg/m^2^
3, 44	Etoposide 100 mg/m^2^
21, 63	Cyclophosphamide 1500 mg/m^2^
	Etoposide 100 mg/m^2^
	Vincristine 1.5 mg/m^2^
22, 23, 64, 65	Etoposide 100 mg/m^2^
**Radiotherapy**	
Day	Therapy
84	25 Gy craniospinal, 55 Gy to tumor bed
**Consolidation**	
Day	Therapy
133, 140, 147	Vincristine 1.5 mg/m^2^
154, 196, 238, 280, 322, 364, 406, 448	CCNU 75 mg/m^2^
	Cisplatin 75 mg/m^2^
	Vincristine 1.5 mg/m^2^

**Table 2 children-10-01387-t002:** High-risk treatment protocol.

Induction	
Day	Therapy
1	Carboplatin 500 mg/m^2^
	Etoposide 100 mg/m^2^
	Vincristine 1.5 mg/m^2^
2	Carboplatin 500 mg/m^2^
	Etoposide 100 mg/m^2^
3	Etoposide 100 mg/m^2^
21–25	Cisplatin 20 mg/m^2^
	Etoposide 60 mg/m^2^
	Ifosfamide 900 mg/m^2^
42	Carboplatin 500 mg/m^2^
	Etoposide 100 mg/m^2^
	Vincristine 1.5 mg/m^2^
43	Carboplatin 500 mg/m^2^
	Etoposide 100 mg/m^2^
44	Etoposide 100 mg/m^2^
63–67	Cisplatin 20 mg/m^2^
	Etoposide 60 mg/m^2^
	Ifosfamide 900 mg/m^2^
**Radiotherapy**	
Day	Therapy
84	36 Gy craniospinal, 55 Gy to tumor bed +/− boost to metastases
**Consolidation**	
Day	Therapy
140, 147, 154	Vincristine 1.5 mg/m^2^
161, 203, 245, 287, 329, 371, 413, 455	Cisplatin 75 mg/m^2^
	CCNU 75 mg/m^2^
	Vincristine 1.5 mg/m^2^

**Table 3 children-10-01387-t003:** Patient characteristics.

	Full Cohort (*N* = 153)	Standard-Risk (*N* = 67)	High-Risk (*N* = 86)
**Sex (%)**			
Male	104 (68)	42 (63)	62 (72)
Female	49 (32)	25 (37)	24 (28)
**Age at diagnosis**			
Median, years (IQR)	8.0 (5.4–11.0)	7.1 (5.1–10.7)	8.4 (5.8–11.0)
Age 0–5 years (%)	29 (19.0)	15 (22.4)	14 (16.3)
Age 5–10 years (%)	78 (51.0)	33 (49.3)	45 (52.3)
Age > 10 years (%)	46 (30.0)	19 (28.3)	27 (31.4)
**Follow-up Time**			
Median, years (IQR)	6.4 (3.6–9.2)	6.8 (3.8–9.9)	6.2 (3.6–9.0)
**Extent of Resection (%)**			
GTR	124 (81)	67 (100)	57 (66)
STR	29 (19)	0 (0)	29 (34)
**Extent of Metastasis (%)**			
M0	53 (35)	55 (82)	7 (8)
M+	63 (41)	0 (0)	54 (63)
Unknown	37 (24)	12 (18)	25 (29)
**Histology (%)**			
Large-Cell Anaplastic	43 (28)	0 (0)	43 (50)
Other	110 (72)	67 (100)	43 (50)
**Molecular Subtype (%)**			
WNT	11 (7.2)	7 (10.4)	4 (4.7)
SHH	2 (1.3)	0 (0)	2 (2.3)
Unknown	140 (91.5)	60 (89.6)	80 (93)
**Gene Amplification (%)**			
MYC	4 (2.6)	1 (1.5)	3 (3.5)
MYCN	3 (2.0)	1 (1.5)	2 (2.3)

**Table 4 children-10-01387-t004:** Survival outcomes by risk group.

	Standard-Risk (*N* = 67)	High-Risk (*N* = 86)
Overall Survival at 5 Years (SE)	87 (4.3)	81 (4.3)
Event-free Survival at 5 Years (SE)	84 (4.6)	79 (4.5)
Disease Recurrence (%)	12 (18)	18 (21)
Alive at Time of Last Follow-up (%)	56 (84)	68 (79)

SE: Standard error.

## Data Availability

The data presented in this study are available by request from the corresponding author.

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
