# Peer review of "Risk-Adapted Treatment Strategies with Pre-Irradiation Chemotherapy in Pediatric Medulloblastoma: Outcomes from the Polish Pediatric Neuro-Oncology Group"

_children, 2023, doi:10.3390/children10081387_

Round 1
Reviewer 1 Report
To the authors:
Your manuscript summarizes the outcomes from the Polish Pediatric Neuro-oncology Group. It shows good survival rates in older patients with medulloblastoma by using a treatment approach that includes pre-irradiation chemotherapy. Your manuscript may help to understand how to reduce toxicity in patients with medulloblastoma who undergo radiation therapy. Attached you will find some major/minor issues regarding your manuscript.
Major/Minor Issues:
1. Please provide the ethical approval number.
2. Line 195-196: Please indicate range of follow-up.
3. Is it possible to evaluate neurotoxicity/ototoxicity and/or cognitive impairment in your patient cohort compared to other treatment regimens?
4. Please indicate and discuss the most recent survival rates from other studies in your discussion.
Good English language.
Reviewer 2 Report
Abstract:
The abstract is very well structured, sufficiently concise, accompanied by 5 keywords that are well chosen and suggestive for the manuscript.
On a scale of 1 to 10, I’ll give 10 points for the abstract.
Introduction:
The introduction is well written, in an interesting and attractive way for the potential reader; 18 bibliographic references support the text very well.
On a scale of 1 to 10, I agree 9 points for introduction.
Methodology:
The methodology is correctly described, sufficiently detailed; I did not notice any notable deficiencies. The 2 tables complete the presentation of the treatment protocol very well. The methodology deserves maximum marks.
On a scale of 1 to 10, I agree 10 points for methodology.
Results:
The results are interesting, quite well presented, without excess text, 1 table and 2 figures are included, but I think that in terms of graphical presentation of the results there is room for improvement.
On a scale of 1 to 10, I agree 8 points for results.
Discussion:
The discussions represent the most consistent part of this manuscript; they are supported by sufficient bibliographic references. The authors do not forget to emphasize the limitations of their study. This chapter seems to be the best of the entire manuscript.
In this situation, on a scale of 1 to 10, I agree 10 points for discussion.
Conclusion:
The conclusions are correctly written, they conclude this manuscript in a good manner. Only one aspect is missing, namely the indication of potential future research directions.
On a scale of 1 to 10, I agree 9 points for conclusions.
Bibliography/References:
38 references, current, correctly written and correctly quoted in the text, represent an acceptable level for such a manuscript.
On a scale of 1 to 10, I agree 9 points for the bibliography.
Figures/Tables:
I identified 2 figures and 3 tables, of good quality, with satisfactory resolution, which are necessary and useful for the manuscript. However, in terms of graphic representation, the manuscript could be improved.
The observation I made in the "Figures/tables" chapter is in close correlation with what I noted in the "Results" chapter. That is, the authors obtained interesting results, which they unfortunately presented mostly in text and quite little in graphic form. Regarding the results, I think that the graphic representation should be improved, because, let's be realistic, a reader will go through a text full of numerical values with a little patience/attention/pleasure, but will be much more receptive to graphic representations, let's say of the chart type , which can illustrate very clearly/well the results obtained by the study in question.
On a scale of 1 to 10, I agree 8 points for this chapter.
Review Decision:
Accept after minor revision.
Round 2
Reviewer 2 Report
The changes made by the authors increase the value of the manuscript. I have no other comments to make, I consider the revised form of the manuscript to be good enough and I recommend its acceptance for publication.